Arterial spin labeling versus BOLD in direct challenge and drug-task interaction pharmacological fMRI

Stewart Stephanie B. 1 2
Koller Jonathan M. 1
Campbell Meghan C. 2 3
Black Kevin J. 1 2 3 4 5 kevin@WUSTL.edu
1 Department of Psychiatry, Washington University School of Medicine , St Louis, MO , USA
2 Department of Neurology, Washington University School of Medicine , St Louis, MO , USA
3 Department of Radiology, Washington University School of Medicine , St Louis, MO , USA
4 Department of Anatomy and Neurobiology, Washington University School of Medicine , St Louis, MO , USA
5 Division of Biology and Biomedical Sciences, Washington University School of Medicine , St Louis, MO , USA
Reser David
Electronic publication date: 2014 Dec 11
Publication date: 2014
Volume: 2
Electronic Location ID: e687
Received 2014 Sep 9; Accepted 2014 Nov 16
Copyright: © 2014 Stewart et al.
Copyright year: 2014
Copyright holder: Stewart et al.
License: This is an open access article distributed under the terms of the Creative Commons Attribution License, which permits unrestricted use, distribution, reproduction and adaptation in any medium and for any purpose provided that it is properly attributed. For attribution, the original author(s), title, publication source (PeerJ) and either DOI or URL of the article must be cited.
License URL: https://creativecommons.org/licenses/by/4.0/

Keywords: phMRI (pharmacological fMRI), Functional magnetic resonance imaging, Pulsed arterial spin labeling, Tozadenant, Statistical parametric mapping, Arterial spin labeling (ASL), BOLD, Parkinson disease

Funding: National Institutes of Health C06 RR020092 UL1 RR024992 P30 NS048056 U54 CA136398-02900209 K24 MH087913 T32 DA007261 American Parkinson Disease Association (APDA) Advanced Research Center Greater St. Louis Chapter The original data collection was supported by a Synosia Therapeutics contract to Washington University (PI: KJB), with infrastructure support from the National Institutes of Health (NIH; grants C06 RR020092, UL1 RR024992, P30 NS048056, U54 CA136398-02900209), the American Parkinson Disease Association (APDA) Advanced Research Center for Parkinson Disease at Washington University, and the Greater St. Louis Chapter of the APDA. The current analysis was supported by NIH grants K24 MH087913 and T32 DA007261. The funders had no role in study design, data collection and analysis, decision to publish, or preparation of the manuscript.

==============================
A carefully controlled study allowed us to compare the sensitivity of ASL (arterial spin labeling) and BOLD (blood oxygen level dependent) fMRI for detecting the effects of the adenosine A2a antagonist tozadenant in Parkinson disease. The study compared the effect of drug directly or the interaction of the drug with a cognitive task. Only ASL detected the direct effect of tozadenant. BOLD was more sensitive to the cognitive task, which (unlike most drugs) allows on–off comparisons over short periods of time. Neither ASL nor BOLD could detect a cognitive-pharmacological interaction. These results are consistent with the known relative advantages of each fMRI method, and suggest that for drug development, directly imaging pharmacodynamic effects with ASL may have advantages over cognitive-pharmacological interaction BOLD, which has hitherto been the more common approach to pharmacological fMRI.

Introduction

Pharmacological magnetic resonance imaging (phMRI) uses fMRI to determine drug-induced changes in brain activity and has multiple applications for pharmaceutical development and efficacy testing. Before the development of functional MRI (fMRI), pharmacological brain imaging most often directly compared brain activity on drug to brain activity off drug (Herscovitch, 2001; McCulloch, 1982). Generally, phMRI studies have avoided this direct approach. Some used drugs with rapid onset and rapid decay of action, and correlated brain BOLD (blood oxygen level dependent) signal with noticeable transient physiological effects, e.g., repeated ratings of cocaine-induced “high” (Breiter et al., 1997). Other phMRI studies used drugs with rapid uptake and rapid elimination, with sequential measurements of plasma concentration, to detect brain changes with the expected pharmacokinetics (Bloom et al., 1999). Drug effects on functional connectivity have also been examined (Schwarz et al., 2007). The most common phMRI approach examines the interactive effects of a drug on the BOLD signal changes induced by a cognitive or sensory stimulus (Cole, Schwarz & Schmidt, 2012; Moeller et al., 2012; Wise et al., 2002). All of these study designs were motivated in part by limitations of BOLD fMRI, whose signal is nonquantitative and fluctuates artifactually over space and time (Iannetti & Wise, 2007).

By contrast, ASL (arterial spin labeling) is an fMRI method that produces a temporally stable signal. Additionally, ASL images reflect regional cerebral blood flow (rCBF) and thus allow relatively straightforward physiological interpretation. These advantages have led some recent drug discovery phMRI studies to use ASL (for reviews, see: Wang et al., 2011; Zelaya et al., in press). Citalopram (Chen et al., 2011) and amphetamine (Nordin et al., 2013) are two examples of psychoactive drugs studied using ASL.

These considerations, and our experience with pharmacological challenge positron emission tomography (PET) blood flow imaging (e.g., Black et al., 2002; Hershey et al., 1998), led us to choose a pure pharmacological challenge approach with perfusion fMRI for a pharmacological challenge MRI study in Parkinson disease (Black et al., 2010b). However, we designed the study so that we would also have data from the more prevalent BOLD drug-task interaction design. The resulting data set allows a fair comparison of these two methods, i.e., subjects provided imaging data for both methods during the same imaging sessions, with similar drug concentrations, the same task, and similar total MRI acquisition times. While previous studies have used ASL for phMRI, to our knowledge this is the first direct comparison of ASL and BOLD for phMRI.

Materials & Methods

Study participants

Fourteen nondemented, nondepressed, ambulatory adults age 40–75, 11 men, with idiopathic Parkinson disease, Hoehn & Yahr stage 1–3 (mean stage 2) (Hoehn & Yahr, 1967), treated with a stable dose of levodopa but no dopamine agonists, participated in the clinical trial (registered at http://clinicaltrials.gov with identifier NCT00605553). Detailed inclusion and exclusion criteria were reported previously (Black et al., 2010a). The study was approved by the Washington University Human Research Protection Office (IRB), approval # 08-0059, and all subjects provided written documentation of informed consent prior to participation.

Study protocol

Subjects were randomly assigned to one of two treatment groups: one group took 60 mg of the adenosine A2a antagonist tozadenant (SYN115) twice daily for one week, waited for a one week washout period and then took a matching placebo twice daily for one week; those assigned to the other group participated in the reverse order. The original report included additional subjects allocated to 20 mg vs placebo, but for this report we focus only on the 60 mg arms. Adenosine A2a antagonists have been studied eagerly as potential treatments for Parkinson disease, alone or in combination with standard antiparkinsonian therapy (Pinna, 2014). A2a receptors occur together with dopamine D2 and D3 receptors on striatopallidal neurons that inhibit the indirect basal ganglia pathway, and A2a antagonists mimic some of the biological effects of dopamine D2 and D3 agonists (reviewed in Black et al., 2010b).

Subjects and investigators were blind to the group assignments. Neuroimaging was performed on the last day of each treatment week. On the morning of the scan day, they did not take their usual antiparkinsonian medications, but did take the last dose of tozadenant or placebo at approximately 6:00 AM. The timing of the fMRI assessments was planned to approximately bracket the time to maximal plasma concentration of tozadenant after chronic dosing. Subjects took 200 mg of carbidopa on arrival to the imaging center and then underwent two sets of MRI assessments, once before and once during an infusion of levodopa, dosed to produce a steady plasma concentration of 600 ng/mL. Levodopa is a precursor to dopamine and is used in Parkinson’s disease to ameliorate the deficiency of dopamine in the substantia nigra. The carbidopa pretreatment was given to inhibit peripheral metabolism of the upcoming levodopa infusion, minimizing side effects from dopamine production in the periphery and keeping more of the levodopa available to the brain.

Subject behavior

Each scanning session included two perfusion MRI (ASL) runs while the subject performed the 2-back memory task, two control ASL runs while the subject fixated on a crosshair, and two block-design BOLD runs, each with four fixation blocks bracketing three task blocks (Fig. 1). ASL scans were also obtained for additional tasks without a BOLD comparison. In each session the fixation ASL and 2-back ASL scans were acquired immediately after the BOLD runs. The 2-back task inter-stimulus interval was 2.5 s for both ASL and BOLD.

Figure 1 Scan day study design.

Each BOLD run comprised 10 s fixation, 50 s 2-back, 30 s fixation, 50 s 2-back, 30 s fixation, 50 s 2-back, and 40 s fixation.

This study employed a working memory task for several reasons. Working memory performance is affected by Parkinson disease and is sensitive to manipulations of central dopaminergic transmission (Cools & D’Esposito, 2011; Hershey et al., 2004). Adenosine A2a receptor antagonists interact with dopamine receptors and can improve working memory performance (Takahashi, Pamplona & Prediger, 2008), including in animal models of parkinsonism (Kadowaki Horita et al., 2013). Several cognitive-pharmacological interaction phMRI studies have employed working memory tasks (Barch et al., 2012), including another study with tozadenant (Moeller et al., 2012). For these and other reasons, several A2a antagonists have been studied for possible cognitive benefits in PD (Pinna, 2014).

One subject was excluded from all analyses presented here because his 2-back task performance was less than 70% accurate. All other subjects had greater than 70% on every run. We previously reported that tozadenant at this dose had no statistically significant effect on 2-back performance, including accuracy and response time (Campbell et al., 2010).

MR image acquisition

All MRI data were acquired at 3T on the Siemens Magnetron Tim Trio with the 12-channel matrix head coil. BOLD-sensitive echo-planar images (EPI) were obtained with flip angle 90°, echo time (TE) 27 ms, repetition time (TR) 2000 ms, 36 planes with interleaved slice acquisition, field of view (256 mm)2, and voxel size (4.0 mm)3. Over a period of 4.33 min for each run, 130 volumes (frames) were acquired; the first 4 frames were discarded to ensure steady-state magnetization.

ASL images were acquired with the commercial Siemens pulsed ASL (pASL) sequence (Wang et al., 2003b). Fifteen transverse echo-planar readout slices with center-to-center slice distance 7.5 mm were acquired with (64)2 (3.4375 mm)2 voxels in each plane, TE 13.0 ms, TR 2600 ms, and flip angle 90°. Labeling slab thickness was 10 cm. Fat suppression was used. The perfusion mode was PICORE Q2T, with TI1 700 ms, saturation stop time 1600 ms and TI2 1800 ms. An M0 image was followed by 31 tag–control pairs for a total acquisition time for each ASL run of 2.73 min.

Brain structure was assessed from sagittal magnetization-prepared rapid gradient-echo (MP RAGE) acquisitions with voxel size (1.0 mm)3, TE = 3.08 ms, TR = 2400 ms, TI = 1,000 ms, flip angle = 8° (Mugler III & Brookeman, 1990), one at each of the 4 scanning sessions. The structural images for each subject were inspected visually, images of lower quality were rejected, and the remaining 1-4 MP-RAGE images for each subject were mutually registered and averaged using a validated method (Black et al., 2001).

Image preprocessing

BOLD images from each subject were preprocessed to reduce artifacts, including correction for intensity differences due to interleaved acquisition, interpolation for slice time correction, correction for head movement, and alignment to atlas space (Hershey et al., 2004). Image intensity was adjusted on a frame-by-frame basis so that each frame had a whole-brain modal value of 1,000 (Ojemann et al., 1997). Frames were smoothed using a 6 mm (FWHM) Gaussian filter and resampled to (3 mm)3 cubic voxels. To minimize motion-related artifact, frames were removed if framewise displacement exceeded 0.9 mm (Siegel et al., 2014).

The 63 frames of the ASL run were smoothed using a 5.7 mm (FWHM) Gaussian filter (resolution chosen to best match the final smoothing estimated from the BOLD images) and rigidly aligned using a method validated in humans and other species (Black et al., 2001; Black et al., 2014). Cerebral blood flow (CBF) was computed in each voxel for each tag-control EPI pair as described (Wang et al., 2003b). The aligned EPI images were also summed to facilitate later alignment steps, and the summed, aligned EPI images from each run were mutually aligned within each subject and summed across runs. The resulting summed EPI images from each subject were affine registered to a target image in Talairach and Tournoux space made using validated methods from these subjects’ structural MR images (Hershey et al., 2004). The products of the registration matrix from this step and the matrices from the within-run mutual registration step were used to resample the 31 tag–control pair CBF images from each run into atlas space images with (3 mm)3 cubic voxels in a single resampling step. As with the BOLD, to minimize motion-related artifact we removed tag–control pairs if framewise displacement in either EPI image exceeded 0.9 mm (Siegel et al., 2014). One subject’s data was excluded from further analysis because over half of his frame pairs were removed due to head motion, leaving 12 subjects for all analyses below. The CBF images in atlas space from the remaining pairs were averaged to create one atlas-registered CBF image for each ASL run. Each CBF image was scaled to a modal global (whole-brain) CBF of 50 mL/hg/min (Stewart et al., 2014). See Fig. 2 for an example CBF image.

Figure 2 ASL blood flow image of one subject’s 2-back run before levodopa on the placebo day.

Statistical analysis

Analysis strategy

The analyses were designed so that each ASL–BOLD comparison included the same scan sessions from the same group of subjects, and as nearly as possible the same image smoothness. Furthermore, the images used to compare the modalities were t images from the same sample, and hence were commensurate. Statistical images were created for each imaging modality to examine the 2-back task effect, the interaction of the 2-back task with tozadenant, and the direct comparison of tozadenant versus placebo.

Statistical images

To identify regions of activation and deactivation, we used a mixed-effects approach with partitioned variance (Penny & Henson, 2007). First, for each study subject, we used a voxelwise general linear model (GLM) that included main effects of task (2-back vs. fixation), levodopa (during vs. before infusion) and drug (tozadenant vs. placebo), and their interactions. For each effect analyzed (2-back task, drug-task interaction, drug effect), SPM12b software (www.fil.ion.ucl.ac.uk/spm/) generated a contrast image for each subject from ASL data, and fIDL (http://www.nil.wustl.edu/~fidl/) did the same for BOLD images (also correcting for linear drift within each run). Note for each subject, every contrast image for ASL data was derived from the same set of scans, and similarly for the BOLD data. These single-subject contrast images (t images) were used as input to second-level statistical parametric mapping (SPM) analyses based on a voxelwise GLM with a covariate for subject age and a factor for sex. At each voxel, GLM contrasts generated t images to test whether the single-subject contrast images at that voxel were significantly less than or greater than zero, across subjects, taking age and sex into account. After thresholding at the t value corresponding to uncorrected p = .001, multiple comparisons correction was performed with the cluster false discovery rate set at p = .05. Approximate anatomical locations of peaks in the statistical images were provided by the Talairach Daemon client (www.talairach.org) (Lancaster et al., 1997; Lancaster et al., 2000).

Secondary analysis: effects of levodopa

The study design was optimized for tozadenant rather than levodopa, and the levodopa dose was relatively low, so analyses examining the effect of levodopa were secondary. To investigate the effects of levodopa we created statistical images of the levodopa effect (comparing scans acquired during the levodopa infusion to scans prior to infusion), of the interaction of the 2-back task with levodopa, and of the 3-way interaction of the 2-back task, levodopa and tozadenant.

Results

Cross-modality image comparison

The final resolution of the (3 mm)3 ASL and BOLD images was similar (Table 1). Total acquisition time was about 25% longer for ASL than BOLD, but acquisition time for the data actually submitted to statistical analysis was much more similar (Table 1), largely because each head movement lost 5.2 s of data in the ASL data versus 2.0 s in the BOLD data.

Table 1 Comparison of BOLD and ASL images.

	BOLD	ASL	
Total acquisition time per scanning session	8.7 min	10.9 min	
Acquisition time per session, limited to frames
retained after motion censoring (mean ± SD)	8.5 ± 0.1 min	9.2 ± 1.1 min	
FWHM (x × y × z)a	10.1 × 10.5 × 9.0 mm	9.4 × 10.5 × 11 mm	
Notes.

a Average of the FWHM estimates across SPM analyses.

Task activation

The working memory task serves as a positive control, and significant regional activations were identified. The analysis using the ASL data identified one significant activation cluster (22 voxels = 0.6 ml, corrected p = 0.030, peak t = 5.88 at −32, −3, 57, left middle frontal gyrus, Brodmann area [BA] 6) (Fig. S1). The analysis using the BOLD data identified 12 significant clusters; the largest cluster also included −32, −3, 57 (515 voxels = 13.9 ml), corrected p < .001, peak t = 12.29 at −40, 3, 33 (left precentral gyrus, BA6) (see Table 2, Fig. S2A). There were no significant deactivations in the ASL data, while the analysis using the BOLD data identified 11 significant deactivation clusters (the largest had volume 2,142 voxels = 57.8 ml, corrected p < .001, peak t = 12.70 at −4, −54, 12, left posterior cingulate, BA29) (Table 3, Fig. S2B).

Table 2 Significant activations during 2-back task (BOLD).

#	Cluster
volume,
voxels	Cluster
volume,
cm3	p (FDR)	Peak t	Atlas location of
peak t value	Anatomical location
of peak ta	
1	515	13.9	<.001	12.29	−40 3 33	Left precentral gyrus (BA 6)	
2	471	12.7	<.001	9.80	4 12 48	Right superior frontal gyrus (BA 6)	
3	327	8.8	<.001	10.75	56 −54−12	Right inferior temporal gyrus (BA20)	
4	224	6.0	<.001	9.40	−40 −63−24	Cerebellum, left posterior lobe	
5	223	6.0	<.001	8.73	44 27 30	Right middle frontal gyrus (BA9)	
6	166	4.5	<.001	7.53	−10 −18 12	Left caudate	
7	163	4.4	<.001	6.38	44 −48 51	Right postcentral gyrus (BA2)	
8	142	3.8	<.001	13.42	32 21 6	Right insula (BA 13)	
9	127	3.4	<.001	12.94	−28 21 3	Left claustrum	
10	108	2.9	<.001	8.41	−2 −81−27	Left cerebellum	
11	47	1.3	<.001	7.69	−28 −57 42	Left superior parietal lobule (BA7)	
12	22	0.6	.016	6.30	−38 48 18	Left superior frontal gyrus (BA10)	
Notes.

a BA, Brodmann area.

Table 3 Significant deactivations during 2-back task (BOLD).

#	Cluster
volume,
voxels	Cluster
volume,
cm3	p (FDR)	Peak t	Atlas location of
peak t value	Anatomical locationa	
1	2,142	57.8	<.001	12.70	−4 −54 12	Left posterior cingulate (BA29)	
2	507	13.7	<.001	8.03	4 12 0	Right caudate	
3	360	9.7	<.001	7.76	−38 −18 21	Left insula (BA13)	
4	132	3.6	<.001	8.78	−44 −75 30	Left angular gyrus (BA39)	
5	104	2.8	<.001	6.72	52 −75 21	Right middle temporal gyrus (BA19)	
6	65	1.8	<.001	6.81	−56 0 −15	Left middle temporal gyrus (BA21)	
7	59	1.6	<.001	7.57	26 6 −21	Right uncus (BA28)	
8	46	1.2	.001	9.74	10 −51−42	Right cerebellar tonsil	
9	42	1.1	.001	6.50	32 −72−33	Right cerebellum, pyramis	
10	40	1.1	.001	6.68	−34 −18 0	Left lentiform nucleus	
11	29	0.8	.006	7.18	14 39 54	Right superior frontal gyrus (BA8)	
Notes.

a BA, Brodmann area.

Drug effect

The task-drug interaction (tozadenant ×2-back) showed no significant results for ASL or BOLD (Figs. S3 and S4). However, the drug vs. placebo contrast from the same ASL data revealed significant rCBF decreases on tozadenant in the thalamus bilaterally (Table 4, Fig. 3, Fig. S5). There were no significant clusters of increased rCBF. As expected, the same contrast with the BOLD data found no significant clusters of activation or deactivation (Fig. S6). Table 5 summarizes all these contrasts.

Table 4 Significant clusters of decreased rCBF on tozadenant.

#	Cluster volume,
voxels (cm3)	p (FDR)	Peak t	Atlas location	Anatomical location of peak t	
1	25 (0.68)	.004	5.67	8, −15, 9	Right medial dorsal nucleus of thalamus	
2	10 (0.27)	.049	5.17	−8, −21, 9	Left medial dorsal nucleus of thalamus	

Table 5 Summary of activation clusters for all contrasts.

Contrast	Number of significant clusters	
	ASL	BOLD	
2-back activation	1	12	
2-back deactivation	0	11	
Tozadenant × 2-back activation	0	0	
Tozadenant × 2-back deactivation	0	0	
Tozadenant activation	0	0	
Tozadenant deactivation	2	0	

Figure 3 Coronal (A), axial (B) and sagittal (C) sections showing the significant CBF decreases on tozadenant 60 mg twice daily.

Colored voxels indicate p < .001 uncorrected; the corrected p value is .004 for the cluster in right thalamus and .049 for the left thalamus (see also Table 4).

Levodopa effect

There were no significant clusters for the pure levodopa effect (Figs. S7 and S8), the task- levodopa interaction (Figs. S9 and S10), or the 3-way interaction (Figs. S11 and S12) in either the ASL or the BOLD images.

Discussion

Cognitive-pharmacological interaction is a common phMRI approach. However, in this study neither ASL nor BOLD analyses detected significant clusters for the interaction of tozadenant with 2-back task activation, whereas directly comparing rCBF on versus off drug using ASL did reveal significant differences. The drug-induced rCBF decreases detected by ASL are in the thalamus, consistent with animal studies suggesting that adenosine A2a receptor antagonists inhibit neuronal activity in the indirect pathway, including in pallidal afferents to thalamus (Black et al., 2010b).

Although the sample size was modest, positive controls built into the experiment confirm that the absence of significant drug effects in the BOLD analysis cannot comfortably be attributed to inadequate image quality or limited data: these same scans were quite adequate to detect significant cognitive (2-back task) effects in a pattern consistent with previous functional imaging studies on working memory (Barch et al., 2012; Bledowski, Kaiser & Rahm, 2010). BOLD is generally more sensitive than ASL for comparisons like this one that can be made over very brief time intervals (a minute or so) (Wang et al., 2003a). However, noise in BOLD data worsens as the time between activation and control acquisitions increases (Aguirre et al., 2002; Ollinger, Corbetta & Shulman, 2001; Zarahn, Aguirre & D’Esposito, 1997), and this temporal instability likely explains why the BOLD data could not detect direct drug effects between sessions. By contrast, the temporal stability of ASL may suit it better to measure the effects of medications, which after all often have been optimized to require only a few doses a day, and hence have slow onset and wearing off of action (Aguirre et al., 2002; Wang et al., 2011; Zelaya et al., in press). A different solution to BOLD’s limited temporal stability is functional connectivity fMRI with and without drug (Schwarz et al., 2007).

Comparing scans from different sequences was feasible here because both BOLD and ASL data were acquired during the same scan sessions in the same subjects, and because the images submitted to statistical analysis were of similar spatial smoothness. Also, in each scan session, the ASL scans immediately followed the two BOLD runs, so that any slowly evolving effects of practice, fatigue or drug should be similar for the two modalities.

Limitations of this study include the imperfect matching between ASL and BOLD of total acquisition time and original voxel size. The different original voxel size is in part a technical limitation because ASL is best suited to acquiring read-out planes in inferior-to-superior order, whereas BOLD can be acquired with even and odd read-out planes interleaved. We used an early version of this pASL sequence, and newer ASL sequences may be even more sensitive to pharmacological agents (Alsop et al., in press). Additionally, most of the subjects in this sample are male, consistent with the higher prevalence of Parkinson disease in men. However, sex differences likely are irrelevant to the comparison of BOLD and ASL.

These were the first Parkinson disease patients ever to receive the drug, so ideal dosing was not yet known. In fact, the initial imaging results from this study suggested that higher doses might be more effective (Black et al., 2010b). Thus the later phase 2b study included higher doses of tozadenant, and demonstrated significant reductions in mean daily “off” time at 120 or 180 mg twice daily but not at 60 mg twice daily (Hauser et al., 2014). Thus another limitation of the present study is that more robust phMRI results may have been found with a higher dose of drug. Nevertheless, tozadenant at 60 mg twice daily did improve tapping speed compared to placebo, whether on or off levodopa (Black et al., 2010a). More importantly, early studies with a new compound most appropriately begin with low doses, and the drug challenge ASL approach was able to detect alterations in brain activity even at these relatively low doses.

One additional advantage of this study comes from the following consideration. A drug that produces symptomatic effects, for instance a feeling of alertness, may cause secondary effects on neuronal activity via the effect on emotional state in addition to any direct neuronal effects (including the neuronal effects that themselves produce the sense of alertness). The same reasoning applies to any placebo effect that may be heightened if the subject notices any drug effect. In this study, most subjects were unable to distinguish whether they were taking the active drug or the placebo, allowing more straightforward interpretation of the drug’s effects on neuronal activity.

Decreased thalamic rCBF with tozadenant was also the most significant result of the previously published analysis of ASL data from this study (Black et al., 2010b). The present analysis detected fewer significant voxels, but several factors account for the difference. In order to match the BOLD data, the present analysis excluded half the ASL data (acquired during additional behavior states for which there were no comparable BOLD data) and smoothed the data less than in the published analysis. The current analysis also excluded subjects with excessive movement or poor 2-back task performance, censored frames for head motion, and improved the correction for global CBF.

Despite the small size and low dose, ASL was sensitive enough to capture phMRI activity. While BOLD may be able to capture task-drug interaction or direct pharmacological effects with larger sample sizes or higher doses, early pharmacological studies are more feasible in smaller samples using lower doses. In summary, these data offer direct, head-to-head evidence using a cognitive task that phMRI using ASL and pure pharmacologic activation may be more sensitive than task-drug-interaction BOLD phMRI, especially for early stage phMRI studies.

Supplemental Information

Supplemental Information 1 SPM of SYN effect using ASL

Click here for additional data file.

Supplemental Information 2 SPM of SYN effect

Click here for additional data file.

Supplemental Information 3 SPM of 2 back effect using ASL

Click here for additional data file.

Supplemental Information 4 SPM of 2 back effect using BOLD

Click here for additional data file.

Supplemental Information 5 SPM of SYN × 2 back effect using ASL

Click here for additional data file.

Supplemental Information 6 SPM of SYN × 2 back effect using BOLD

Click here for additional data file.

Figure S1 SPM of 2-back effect using ASL

First page shows activation clusters and second page shows no statistically significant deactivation clusters.

Click here for additional data file.

Figure S2 SPM of 2-back effect using BOLD

First three pages show activation clusters and last three pages show deactivation clusters.

Click here for additional data file.

Figure S3 SPM of tozadenant and 2-back interaction using ASL

First page shows no statistically significant activation clusters and second page shows no statistically significant deactivation clusters.

Click here for additional data file.

Figure S4 SPM of tozadenant and 2-back interaction using BOLD

First page shows no statistically significant activation clusters and second page shows no statistically significant deactivation clusters.

Click here for additional data file.

Figure S5 SPM of tozadenant effect using ASL

First page shows no statistically significant activation clusters and second page shows two statistically significant deactivation clusters.

Click here for additional data file.

Figure S6 SPM of tozadenant effect using BOLD

First page shows no statistically significant activation clusters and second page shows no statistically significant deactivation clusters.

Click here for additional data file.

Figure S7 SPM of levodopa effect using ASL

First page shows no statistically significant activation clusters and second page shows no statistically significant deactivation clusters.

Click here for additional data file.

Figure S8 SPM of levodopa effect using BOLD

First page shows no statistically significant activation clusters and second page shows no statistically significant deactivation clusters.

Click here for additional data file.

Figure S9 SPM of levodopa and 2-back interaction using ASL

First page shows no statistically significant activation clusters and second page shows no statistically significant deactivation clusters.

Click here for additional data file.

Figure S10 SPM of levodopa and 2-back interaction using BOLD

First page shows no statistically significant activation clusters and second page shows no statistically significant deactivation clusters.

Click here for additional data file.

Figure S11 SPM of levodopa × tozadenant × 2-back interaction using ASL

First page shows no statistically significant activation clusters and second page shows no statistically significant deactivation clusters.

Click here for additional data file.

Figure S12 SPM of levodopa × tozadenant × 2-back interaction using BOLD

First page shows no statistically significant activation clusters and second page shows no statistically significant deactivation clusters.

Click here for additional data file.

Additional Information and Declarations

Competing Interests

Author Contributions

Human Ethics

Kevin J. Black is an Academic Editor for PeerJ.

Stephanie B. Stewart analyzed the data, wrote the paper, prepared figures and/or tables, reviewed drafts of the paper.

Jonathan M. Koller performed the experiments, analyzed the data, reviewed drafts of the paper.

Meghan C. Campbell performed the experiments, reviewed drafts of the paper.

Kevin J. Black conceived and designed the experiments, performed the experiments, analyzed the data, wrote the paper, reviewed drafts of the paper.

The following information was supplied relating to ethical approvals (i.e., approving body and any reference numbers):

Washington University’s Human Research Protection Office, approval # 08-0059.

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
