# Peer review of "Arterial spin labeling versus BOLD in direct challenge and drug-task interaction pharmacological fMRI"

_PeerJ, doi:10.7717/peerj.687_

## Round 0.1 · original submission · Major Revisions

· Academic Editor

Major Revisions

Both reviewers agreed that this is a useful addition to the literature, and I agree, with the proviso that each of the issues identified by the reviewers is carefully addressed in the revised version. If you decide to submit a revision, please identify edited passages which address the reviewers concerns in the text, i.e., using bold or italic fonts. Please also note that Reviewer 2 was kind enough to upload an annotated document, which should be of assistance in your re-working of the manuscript. I look forward to receiving your re-submission.

Reviewer 1 ·

Basic reporting

In the manuscript "Arterial spin labeling versus BOLD in pharmacological fMRI" the authors Stewart et al compare the use of ASL perfusion MRI with task-based BOLD fMRI to detect cerebral physiological changes in a pharmacological setting. The authors show that ASL is more sensitive to detect these changes, most probably due to improved stability over time. This is a valid study, although the authors should make it more explicit in the title, abstract and manuscript that their study only involved task-based BOLD fMRI. Resting state BOLD fMRI is a fundamentally more promising technique for pharmacological MRI than task based BOLD fMRI, but this was not studied by the authors.
Furthermore, the manuscript uses many self-citations, whereas pharmacological ASL studies from other groups are not mentioned or discussed. The authors should objectify their references. Also the use of a slightly outdated ASL sequence (see recommendations paper of Alsop et al in Magn Reson Med) should be mentioned.

Experimental design

The authors fail to provide sufficient information on the employed ASL technique: what PASL-labeling approach was used? What was the width of the labeling slab. What was the inversion time and QUIPSS-timings? Was background suppression employed?

Validity of the findings

The authors should show a single subject CBF-map, so that the reader can judge the image quality.

Additional comments

No comments

·

Basic reporting

The manuscript is written in a clear English. The organization does not follow strictly the template of having the "Results" and "Discussion" sections combined but instead they are separated which, nonetheless, seems to be fine.

Regarding the abstract the authors mention the advantage of using ASL over BOLD in phMRI. Nonetheless, other studies have already used ASL for the same purpose. State clearly in the abstract if this the first time that such comparison between ASL and BOLD is made for phMRI.

In the introduction the authors describe the phMRI approaches and the motivation for the comparison between BOLD and ASL techniques in the presented phMRI study. Nonetheless, I feel that the manuscript is missing the context of why tozadenant was studied in the first place, what is its mechanism of action (it is mentioned that it is an adenosine A2a antagonist but it does not mention which brain regions contain these receptors nor what is the expected effect on Parkison's Disease). This results in a unclear comprehension of why a memory task was used in the study. This fact also reflects a lack of depth in the discussion regarding clinical correlations of the findings (do the results make sense from the physiological point-of-view?). In the discussion only this sentence "adenosine A2a receptor antagonists inhibit neuronal activity in the indirect pathway, including in pallidal afferents to thalamus (Black et al., 2010b)" is mentioned in this regard. It helps to make sense of the observed thalamic changes but I missed the explanation or suggestion of it to why no changes are observed in the other structures of the indirect pathway. Also, no explanation is given to the lack of changes observed in the memory tasks. This is main point of concern for the paper: it is lacking depth in the physiological interpretation and therefore is not translating an important message that the paper may offer.

(In the informed consent is described the why of this study being done. Why not import some content to the paper?)

Also, if a sentence concerning levodopa is included, briefly explaining how it is used in Parkison's Disease, it may help the paper comprehension by the non-specialist in the topic.

I also feel that the results should be further explored in terms of presentation. The authors show only a figure which is supplementary figure 1. Why supplementary? Why not include it in the body of the manuscript. Also, I believe the comprehension of the manuscript would improve if you include other relevant SPM maps.

Regarding references, in the introduction is mentioned: "These considerations, and our experience with pharmacological PET (positron emission tomography) blood flow imaging (Black et al., 1997; Black et al., 2000; Black et al., 2005; Black et al., 2002; Hershey et al., 2003; Hershey et al., 2000; Hershey et al., 1998), led us to choose a pure pharmacological challenge approach with perfusion fMRI for a pharmacological challenge MRI study in Parkinson disease (Black et al., 2010b)". In this sentence 7 references are mentioned to justify that the authors have experience in pharmacological PET. I would have be convinced by one. Further, this sentence includes about one fifth of all the references of the paper and about half of the references of the authors. Do all the references in the sentence bring an added-value to the understanding of the paragraph or the paper? If so, please further detail the contents of the sentence.

Some references are tables need to be formatted.

Supporting files are provided, namely SPM maps, but no explanation is given that would help to understand them. Also, no mention of these maps is made in the text of the manuscript, whilst they may be of importance for readers.

Experimental design

As mentioned above, since no sufficiently detailed explanation regarding tozadenant was provided, it is not understandable why memory tasks were used. Also the authors do not describe other potential tasks to use. Since Parkison's Disease hallmark are motor function impairments why not use a motor task to evaluate the effects of the drug?

The gender characterization of the study sample is missing as well as the subject assessment regarding Parkison's Disease (e.g. using UPDRS) and specifically memory (after all, a memory task was used).

Also, no explanation was provided regarding dosages of 60 mg p.o. bid for 1 week. Hauser et al, Tozadenant (SYN115) in patients with Parkinson's disease who have motor fluctuations on levodopa: a phase 2b, double-blind, randomised trial, Lancet Neurology 13(8) 767-776, 2014 claim that: "Tozadenant 60 mg twice daily was not associated with a significant reduction in off-time" for 12 weeks. Would the authors like to comment on the dosing scheme used regarding the findings of Hauser et al? would this explain the absence of changes observed in some of the examinations/tasks statistical tests?

Please explain the reason for carbidopa administration prior to MRI scans.

Please explain more clearly the goals for each examination/task and also add a schematic of the paradigm design including scanning times for clarity. Also describe how performance of the 2-task was evaluated and present results concerning this evaluation.

ASL images are said to be: "rigidly aligned using a validated method (Black et al, 2001a)". The method described in the paper relates to non-human primates. Could you please comment why the validation holds for humans?

Why was not SPM also used for BOLD images?

Be aware: sometimes CBF acronym is used others rCBF. Please uniformize and/or state differences where appropriate.

In the results section, the estimated FWHM values are mentioned (see table 1) but nowhere is mentioned how these estimations were made. Please include relevant information in the methods section and explain why the smoothness assessment was done.

In the discussion is mentioned: "the present analysis excluded half the ASL data (acquired during additional behavior states for which were no comparable BOLD data) and smoothed the data less than in the published analysis". Why is this not mentioned in the methods and results sections?

In the supplementary material explain what is 3-way interaction.

Validity of the findings

The study sample is quite small: 14 patients. Would this also be an explanation for the obtained results (absence of changes observed in some statistical tests)? In the discussion the authors mention: "Positive controls built into the experiment confirm that the absence of significant drug effects in the BOLD analysis cannot be comfortably
attributed to inadequate image quality or limited data: these same scans were quite adequate to detect significant cognitive (2-back task) effects in a pattern consistent with previous functional imaging studies on working memory". That may be true for a memory task study alone but does this hold for a drug x memory task, where changes may be more suitable and therefore require a larger sample?

As mentioned above I feel that a deeper discussion is missing regarding the correlation of findings with the Parkinson's Disease physiology and the drug mechanism of action. Such discussion would greatly improve the relevance of the paper.

In the conclusion the authors state: "In summary, these data offer direct, head-to-head evidence that phMRI using ASL and pure pharmacologic activation may be more sensitive than task interaction BOLD phMRI". I feel this statement is quite general and arguably supported by the results and discussion. The authors used a memory task in this study, would this statement hold if a motor task was used instead? Please improve manuscript such that conclusions are clearly supported.

Additional comments

The paper is interesting and worth of publication. Nonetheless, as I mentioned above, I feel that some issues should be addressed before publication and that the manuscript clarity should be substantially improved. Additionally, I am sending the annotated manuscript containing additional/specific suggestions and comments.

---

## Round 0.2 · Minor Revisions

· Academic Editor

Minor Revisions

Thank you for the much improved version, and I look forward to seeing the published version of this work. Please carefully review and attend to the suggestions of the reviewers, both of whom have provided detailed suggestions for the minor amendments. It would be helpful to identify amended text by highlighting or italics, to speed the final editorial review of this manuscript.

Reviewer 1 ·

Basic reporting

See first review. The authors have addressed all raised issues.

Experimental design

See first review. The authors have addressed all raised issues.

Validity of the findings

See first review. The authors have addressed all raised issues.

Additional comments

There is one typo: Page 6 ", saturation stop time 1600 mg"should be ", saturation stop time 1600 ms"

·

Basic reporting

see general comments below

Experimental design

see general comments below

Validity of the findings

see general comments below

Additional comments

Dear authors,
The manuscript has greatly improved. I send you the commented manuscript with some minor notes and suggestions for correction. These include in particular:
- some sentences are repeated in the text, as well as a reference is repeated in the references' section;
- there is an isolated reference in the text (Moeller et 2012b), probably belongs to the previous paragraph?
- in the introduction there is a "telegraphic" sentence that I suggest to be further developed;
- state if the mismatch between the number men and women (11 vs 3) limits the validity of the analysis; if not, why?
- clarify MRI acquisition protocol and protocol parameters;
- 2 references have in DOI a website link, why?
- a section of the results seems to be repeated just at the end of the manuscript after the references;
- other minor formal corrections needed.
Congratulations for the interesting work!

---

## Round 0.3 · accepted · Accept

· Academic Editor

Accept

Thank you for your prompt efforts to address the reviewers' concerns, and I look forward to the final publication. I believe that J. Thai (Head of Publishing Operations) has addressed the issue of inadvertent duplication of certain text sections in the upload process, but please check the proof copies carefully for any recurrence of this issue. Thank you again for choosing PeerJ to submit your work.